# Heterodimerization of Mu Opioid Receptor Protomer with Dopamine D_2_ Receptor Modulates Agonist-Induced Internalization of Mu Opioid Receptor

**DOI:** 10.3390/biom9080368

**Published:** 2019-08-14

**Authors:** Lakshmi Vasudevan, Dasiel O. Borroto-Escuela, Jelle Huysentruyt, Kjell Fuxe, Deepak K. Saini, Christophe Stove

**Affiliations:** 1Laboratory of Toxicology, Department of Bioanalysis, Faculty of Pharmaceutical Sciences, Ghent University, 9000 Ghent, Belgium; 2Department of Molecular Reproduction, Development and Genetics, Indian Institute of Science, Bangalore 560012, India; 3Department of Neuroscience, Karolinska Institutet, 17177 Stockholm, Sweden

**Keywords:** G protein-coupled receptor, heterodimerization, mu opioid receptor, dopamine D_2_ receptor

## Abstract

The interplay between the dopamine (DA) and opioid systems in the brain is known to modulate the additive effects of substances of abuse. On one hand, opioids serve mankind by their analgesic properties, which are mediated via the mu opioid receptor (MOR), a Class A G protein-coupled receptor (GPCR), but on the other hand, they pose a potential threat by causing undesired side effects such as tolerance and dependence, for which the exact molecular mechanism is still unknown. Using human embryonic kidney 293T (HEK 293T) and HeLa cells transfected with MOR and the dopamine D_2_ receptor (D_2_R), we demonstrate that these receptors heterodimerize, using an array of biochemical and biophysical techniques such as coimmunoprecipitation (co-IP), bioluminescence resonance energy transfer (BRET^1^), Fӧrster resonance energy transfer (FRET), and functional complementation of a split luciferase. Furthermore, live cell imaging revealed that D_2L_R, when coexpressed with MOR, slowed down internalization of MOR, following activation with the MOR agonist [D-Ala2, N-MePhe4, Gly-ol]-enkephalin (DAMGO).

## 1. Introduction

For hundreds of years, opioids have been used in the management of pain, mediating their analgesic effect primarily via binding to the mu opioid receptor (MOR) [1,2,3]. Dependence and rapid development of tolerance limit the long-term use of opioids [4]. MOR, a member of the Class A G protein-coupled receptor (GPCR) subfamily, is activated by both endogenous opioid peptides as well as exogenous opioids [5]. The latter constitute the potent analgesics, which carry the risk of being used as substances of abuse [6,7]. Although the mechanism of tolerance, defined as the decline in effect of a drug due to its chronic exposure, is not known exactly, it has been linked to receptor desensitization and recruitment of β-arrestin in vivo [8,9,10,11,12]. GPCR kinases (GRKs) phosphorylate the activated receptors, which in turn recruit β-arrestin 1 and β-arrestin 2, and the docking of these proteins shuts off the signal through the G proteins, ultimately resulting in receptor endocytosis [13,14]. In contrast, there are numerous studies that demonstrated that in highly tolerant animals, there was no change in expression of MOR in response to morphine, but this observation has been argued to depend on the dosage, route, and model organism [9,15]. Thus, it could be postulated that there may be other, or a combination of, mechanisms that could affect receptor desensitization and downregulation. 

The dopamine receptor (D_2_R), also a member of the Class A GPCR subfamily, signals upon binding of its catecholamine neurotransmitter, dopamine. Like MOR, D_2_R is also coupled to the Gi/o subunit that inhibits adenylyl cyclase and decreases the production of the second messenger cyclic adenosine monophosphate (cAMP ) [16,17]. D_2_R also causes intracellular Ca^+2^ release through the Gβγ subunits [17]. Several in vitro studies have shown that dopamine receptors undergo phosphorylation by GRKs and recruit β-arrestins in response to agonists, similar to other GPCRs [14,17]. 

Interactions between the opioid and dopamine receptor systems have been shown in several regions of the brain, such as the ventral tegmental area (VTA) and the nucleus accumbens (NAc), which is associated with the reward system and the addictive behavior of drugs [18,19]. Rivera et al. [20] have shown that continuous treatment with morphine caused morphological changes in nigral dopamine nerve cells, which was restored by cotreatment with the dopamine D_4_ receptor (D_4_R) agonist, PD 168,077. Additionally, it was observed that morphine increased locomotion in mice, which was counteracted by the coadministration of PD 168,077. Using a condition place preference paradigm, activation of D_4_R accompanying the administration of morphine was found to decrease the rewarding affects associated with morphine and to attenuate the development of physical dependence associated with morphine. Importantly, the analgesic properties of morphine remained unaltered [20]. A recent study by Dai et al. [21], demonstrated that *levo*-corydalmine (*l*-CDL), a traditional herb used in China to alleviate pain, when coadministered with morphine, attenuated morphine tolerance in mice. *l*-CDL acted as an antagonist of D_2_R and the inhibition of tolerance demonstrated by this compound was reversed upon addition of quinpirole, a D_2_R agonist. Using several pain models in the rat, Mercado-Reyes et al. [22], reported an enhanced antinociceptive effect of the MOR agonist [D-Ala2, N-MePhe4, Gly-ol]-enkephalin DAMGO, when coadministered with quinpirole.

There is substantial evidence for the existence of GPCR dimers, oligomers, and even higher order oligomers. Also, literature suggests a role for heteromerization in modulating receptor functions. Further, D_2_R [23,24] and MOR not only homodimerize [25], but also form heteromers with many other GPCRs. D_2_R has been reported to heterodimerize with the adenosine A_2A_ [26,27], serotonin 5HT_2A_ [28], and cannabinoid CB1 receptor [29,30] whilst MOR heterodimerization with delta opioid receptor (DOR) [31] and cholecystokinin B receptor (CCKBR) [32] has been described. The dopamine D_4_ receptor (D_4_R) has been shown to form heterodimers with both D_2_R and MOR [33,34]. The formation of these heteromers may have an important role in modulating the signaling pathways of the interacting partners, in addition to potentially modifying ligand binding to the receptors. In a study by Dai et al. [35], it was demonstrated that MOR colocalized with D_2_R in the spinal cords of mice and this finding was also confirmed by coimmunoprecipitation. Chronic morphine treatment caused an increase in interaction between the heterodimer, while administration of sulpiride, a D_2_R antagonist, disrupted the interaction between MOR and D_2_R, which led to the attenuation of morphine tolerance, indicating that increased interaction between MOR–D_2_R could have a role in the development of chronic morphine tolerance.

In the present study, we have applied a wide array of complementary techniques, including coimmunopreciptation, bioluminescence resonance energy transfer (BRET^1^), functional complementation (NanoBiT^®^), and Förster resonance energy transfer (FRET) to demonstrate the heterodimerization between MOR and D_2_R. Furthermore, using live cell imaging we observed that the internalization of MOR, when stimulated with its agonist DAMGO, is slowed down in the presence of D_2_R.

## 2. Materials and Methods

### 2.1. Reagents and Antibodies

DAMGO was purchased from Sigma-Aldrich (St. Louis, MO, USA). Dulbecco’s Modified Eagle medium (DMEM), penicillin/streptomycin, glutamine, Trypsin-EDTA (0.05%), Hank’s Balanced Salt Solution (HBSS), Phusion High-Fidelity (HF) PCR Master Mix with HF buffer, Turbofect™ (a transient mammalian cell transfection reagent), Protein A Trisacryl beads, Pierce™ Bicinchoninic acid assay (BCA) Protein Assay Kit, Fluo-4 AM (F14201), and T4 DNA ligase were purchased from Thermo Fisher Scientific (Pittsburg, PA, USA). Fetal bovine serum (FBS) was purchased from Biochrom, which is a now a part of Merck (Merck KGaA, Darmstadt, Germany). Phosphate buffered saline (PBS) was purchased from Lonza (Lonza Walkersville, US). Polyethylenimine (PEI) (a transient mammalian cell transfection reagent), carbenicillin, and Tween 20 were procured from Sigma-Aldrich (Steinheim, Germany). The Nano-Glo^®^ Live Cell reagent was purchased from Promega (Madison, WI, USA). h-coelenterazine was procured from Molecular Probes (Eugene, OR, USA). Primers were synthesized by Eurofins Genomics (Ebersberg, Germany). Restriction enzymes *Hin*dIII, *Eco*RI, and *Xho*I were from New England Biolabs (NEB, MA, USA). Blocking buffer was purchased from LI-COR Biosciences (Lincoln, NE, USA). The antibodies used were: Mouse anti-haemagglutinin (HA) tag (anti-HA16B12) (MMS-101P-1000) from Covance (Princeton, NJ, USA) for co-IP, mouse anti-FLAG^®^ M2 (F3165) from Sigma-Aldrich for co-IP, rabbit anti-GFP (G1544) from Sigma, rabbit anti-HA (GTX29110) from Gene TEX (Irvine, CA, USA), rabbit anti-D_2_R (RRID:AB_2571596) from Frontier Institute (Hokkaido, Japan), goat anti-rabbit IRDye680RD (926-68071), and goat anti-rabbit IRDye800CW (926-32211) from LI-COR Biosciences (Lincoln, NE, USA).

### 2.2. Construction of Plasmids

pHA-D_2L_R was purchased from UMR cDNA Resource Center (www.cdna.org). The plasmid pMOR-*R*Luc was a kind gift from Dr. Francisco Ciruela (University of Barcelona, Barcelona, Spain), pD_2L_R-mCherry was a kind gift from Dr. Ibeth Guevara-Lora (Jagiellonian University, Krakow, Poland), and pmCherry-CAAX was from Dr. Deepak Saini’s laboratory (Indian Institute of Science, Bangalore, India). The plasmids pMOR-YFP, pFLAG-D_2S_R, pD_2L_R-YFP, pD_2L_R-*R*Luc, and pEYFP were kindly provided by Dr. Kjell Fuxe (Karolinska Institutet, Stockholm, Sweden). The plasmid pEGFP was procured from Clontech Laboratories (Saint-Germain-en-Laye, France). The sequences encoding human D_2L_R and MOR were amplified using the primers described in Appendix A using an MJ Research PTC-200 Thermal Cycler (GMI, MN, USA) and cloned in the NanoBiT^®^ system from Promega (Madison, WI, USA). Using this system, the split fragments of nanoluciferase, namely LargeBiT (LgBiT, an 18kD protein) and SmallBiT (SmBiT, an 1kD peptide) were fused C-terminally to the D_2L_R and MOR. HaloTag-SmBiT was also procured from Promega. D_2L_R-EGFP was constructed by digestion of D_2L_R-LgBiT with *Hin*dIII and *Eco*RI and subcloning into the pEGFP-N3 vector (Addgene, Watertown, MA, USA). The constructs containing the appropriate inserts were verified by restriction digest and by sequencing.

### 2.3. Cell Culture and Transfection

HEK 293T (American Type Culture Collection (ATCC), Manassas, VA, USA) cells were used for all heterodimerization studies by co-IP, BRET^1^, and NanoBiT^®^. The cells were cultured in DMEM supplemented with 10% heat inactivated FBS, 2 mM glutamine, 100 IU/mL penicillin, and 100 µg/mL streptomycin and maintained at 37 °C in a humidified atmosphere with 5% CO_2_. For the experiments involving fluorescence microscopy, a human cervical cancer cell line (HeLa) (ATCC, Manassas, VA, USA), maintained under the same conditions as mentioned above, was used. The cells were transiently transfected with relevant plasmids using PEI [36] (co-IP, BRET^1^, NanoBiT^®^), or with Turbofect™ (for imaging studies) and cultured for 48 h.

### 2.4. Coimmunoprecipitation

Cells expressing HA-D_2L_R (or FLAG-D_2S_R) with and without MOR-YFP from a semiconfluent 10 cm^2^ dish, were washed with cold PBS 48 h posttransfection, collected, and frozen at −70 °C prior to adding 300 µl of RIPA buffer (250 mM NaCl; 50 mM Tris/HCl pH 7.5; 1% Nonidet P-40 (NP-40); 0.1% sodium-dodecyl sulfate (SDS), 0.5% deoxycholic acid (fresh) supplemented with protease and phosphatase inhibitors (2.5 µg/mL aprotinin, 1 mM Pefa-block, 10 µg/mL leupeptin, 10 mM β-glycerolphosphate), and mixed at 4 °C for 1 h. An aliquot of the lysate was denatured at 37 °C for 10 min in SDS-sample buffer (4% SDS; 50% glycerol; 0.2% bromophenol blue; 65 mM Tris/HCl pH 6.8 and 50 mM dithiothreitol) and loaded on a 10% SDS-polyacrylamide gel electrophoresis (SDS-PAGE) gel to check the expression of the proteins. The rest of the lysate was incubated with 2 µg of primary antibody (mouse anti-HA16B12 or mouse anti-FLAG^®^M2), while rotating at 4 °C for 4 h. To this, 20 µl of Protein A Trisacryl beads were added and incubated overnight at 4 °C. Subsequently, the sample containing beads was washed three times with RIPA buffer (with inhibitors), denatured at 37 °C for 10 min in SDS-sample buffer, and separated on a 10% SDS-PAGE gel. The separated proteins were transferred onto a nitrocellulose membrane, which was blocked using blocking buffer prior to incubation with the following antibodies, prepared in blocking buffer with TBS (1:1) containing 0.1% Tween 20 (TBS-T): Rabbit anti-HA, anti-GFP, and rabbit anti-D_2_R. Following three washes with TBS-T for 3 times, the blots were incubated with secondary antibodies—goat anti-rabbit antibody coupled with IRDye680RD or goat anti-rabbit antibody coupled with IRDye800CW for 1 h. After 1 h, the blots were washed with TBS-T and imaged with the Odyssey^®^ Infrared Imaging system (IGDR, Rennes, France).

### 2.5. Bioluminescence Resonance Energy Transfer^1^ (BRET^1^)

HEK 293T cells growing on a 6-well plate were transiently transfected with a fixed concentration of donor plasmid (pMOR-*R*Luc or pD_2L_R-*R*Luc) and increasing concentrations of acceptor plasmid (pD_2L_R-YFP or pMOR-YFP or pEYFP), respectively. Then, 48 h post-transfection, the cells were washed twice with warm PBS, detached, centrifuged at 1000× *g* for 10 min, and resuspended in HBSS. An aliquot was used for protein estimation by BCA assay. The cell suspension with a corresponding protein concentration of 600 ng/µL was distributed in duplicate into a black or white 96-well microplate for fluorescence and luminescence measurements, respectively. For luminescence, *h*-coelenterazine at a final concentration of 5 µM was added and measurements were done using a microplate reader, Clariostar (BMG LABTECH, Cary, NC, USA), which allows sequential integration of signals detected at 480 (± 20) nm (luciferase) and 530 (± 20) nm (YFP). The BRET^1^ ratio is expressed as a ratio of the light intensity at 530 nm over the light intensity at 480 nm and is corrected by subtracting the background ratio observed in cells transfected with *R*Luc-tagged receptor alone.

### 2.6. Functional Complementation Assay Using Split Luciferase

HEK 293T cells growing on a 10 cm^2^ dish were transiently transfected with plasmids encoding D_2L_R-LgBiT in combination with MOR-SmBiT or HaloTag-SmBiT, or with MOR-LgBiT in combination with D_2L_R-SmBiT or HaloTag-SmBiT, along with a constant amount of a plasmid encoding enhanced green fluorescent protein (EGFP). Then, 48 h post-transfection, cells were washed twice with warm PBS, detached, centrifuged at 1000× *g* for 10 min, and subsequently suspended in HBSS. Protein estimation was performed on an aliquot using BCA assay. The cell suspensions were diluted to bring all of them to a density corresponding with a protein concentration of 600 ng/µL. Next, 25 µL of Nano-Glo^®^ Live cell reagent containing furimazine substrate (20× diluted using Nano-Glo^®^ LCS dilution buffer) was added to 100 µL of cell suspension, which was added to a 96-well white plate, and the luminescence was measured using the ClarioSTAR. Fluorescence measurements were carried out in a black 96-well plate. The luminescence data in all conditions were normalized to their respective fluorescence signals.

### 2.7. Förster Resonance Energy Transfer (FRET)

HeLa cells growing on a 6-well plate were transiently transfected with plasmids encoding MOR-YFP, combined with either D_2L_R-mCherry or mCherry-CAAX. Then, 48 h post-transfection, the cells were washed with warm PBS, detached and resuspended in HBSS, and an equal number of cells were distributed into a black 96-well plate (Corning Inc., Corning, NY, USA). The fluorescence intensities were recorded in the donor–donor (DD) channel (donor excitation 514/10 nm, donor emission 527/10 nm) and donor–acceptor (DA) channel (donor excitation 514/10 nm, acceptor emission 587/10) using a bottom recording mode in a multimode fluorescence plate reader (Infinite M1000 PRO, Tecan, Austria). The FRET ratio was expressed as a ratio of light intensity in the FRET channel DA divided by light intensity in the donor–donor channel, DD.

### 2.8. Fluorescence Recovery After Photobleaching (FRAP)

FRAP experiments were performed by bleaching a region of interest (ROI) on the plasma membrane using a 488 nm laser set at 100% power with the help of 3i vector system (3i Inc., Denver, CO, USA) on an Olympus IX83 epifluorescence microscope (Olympus, Shinjuku, Japan), connected to a SpectraX fluorescent light source and cascade II EMCCD camera, and controlled using Slidebook 6 software (Intelligent Imaging Innovations, Inc., Denver, CO, USA). Fluorescence recovery was monitored by time lapse imaging in the YFP channel for a total duration of 9 min, with an interval of 10 s between two successive frames. At the 5th frame (i.e., at the 50th second from initial time point 0), the ROIs were photobleached and subsequently, the recovery was measured as described above. FRAP kinetics were calculated as follows: The first step was to normalize the intensity at all time points (I_t_) to the intensity at the first time point of imaging (I_0_, at t = 0 s), thus yielding I_t_/I_0_. In the next step, the intensity at the bleaching time point was subtracted from all the time points, such that the intensity at the time of bleaching was 0 (I_b_). Then, the relative intensities of the individual cells (approximately 50 cells) under each condition (with and without coexpression of D_2L_R) were plotted as a function of time using Graph Pad prism to yield t_1/2_, which is the time required for half maximal recovery after photobleaching.

### 2.9. Fluorescence Live Cell Imaging

HeLa cells growing on a 35 mm glass bottom dish (NEST, Wuxi, China) were transfected with the relevant plasmids and 48 h post-transfection the cells were washed with HBSS containing 10 mM HEPES (pH 7.0) and replaced with HBSS with (for internalization assay) or without calcium (for calcium measurements) for imaging. The fluorescence imaging was performed using an inverted epifluorescence microscope Olympus IX83, equipped with Lumencor Spectra X light engine (Lumencor, Beaverton, OR, USA) and band pass filters in a high-speed filter wheel (ASI Inc., Eugene, OR, USA). The images were acquired using a Cascade II EM-CCD camera (Photometrics Inc., Pittsfield, MA, USA) under controlled temperature and using a CO_2_ incubation system (Okolab, Pozzuoli, Italy). The devices were controlled by Slidebook 6 software, which was also used for data acquisition and analysis.

#### 2.9.1. Measurement of Calcium Release Using Fluo-4 AM by Live Cell Imaging

In cells growing in glass-bottom dishes, the medium was replaced with DMEM (without FBS) containing 1 µM of the calcium binding dye Fluo-4 AM, followed by incubation for 45 min at 37 °C and 5% CO_2_. Next, the cells were washed with calcium free HBSS and imaged in HBSS for 10 min using a 20× objective with a 10 s interval between each image. The ligand was added at the 5th frame (10 s/frame) using a syringe pump. 

#### 2.9.2. Internalization Assay by Live Cell Imaging

Following replacement of the medium with HBSS, images were acquired using a plan-apochromat 60×/1.35 oil-immersion lens, with time lapse imaging. The agonist DAMGO/vehicle was added at the 10th frame (10 s/frame) and cells were monitored for 1 h. All the images were processed by ImageJ software (NIH, Bethesda, MD, USA). Cytosolic intensity was measured by drawing a ROI in each cell, normalized to the basal (unstimulated) condition, and plotted as a function of time.

### 2.10. Statistical Analysis

For all the experiments, n represents independent biological replicates and the data depicted are mean ± standard deviation (SD), unless when mentioned otherwise. The *p* values were calculated using student’s *t*-test with two-tailed distribution.

## 3. Results

### 3.1. Coimmunoprecipitation Indicates that MOR and D_2_R Heterodimerize in Transfected HEK 293T Cells

HEK 293T cells were transiently transfected with plasmids encoding HA-tagged D_2L_R and/or MOR-YFP. Receptor expression was confirmed in total lysates (lanes 5–8 of Figure 1A,B). The presence of HA-D_2L_R in lane 2 of Figure 1B also confirmed that the immunoprecipitation was successful. Immunoprecipitation with anti-HA antibody of HA-tagged D_2L_R in cotransfected HEK 293T cells resulted in the precipitation of YFP-tagged MOR, which was detected using an anti-GFP antibody (lane 4 of Figure 1A). The appearance of bands at a higher molecular weight than predicted indicates the presence of dimers or higher order oligomers that are resistant to denaturation by SDS and may involve hydrophobic interactions [36,37]. As a control, lysates of cells that had been transfected with individual receptors alone were mixed and processed as per the protocol. The lack of co-IP in this experiment confirmed the specificity of the interaction, as well as confirmed that it happens in the same cell only (Appendix A). To gain some insight into the interaction sites, co-IP was also performed with the short isoform of D_2_R (D_2S_R) in a similar way as described above. This is a splice variant of D_2_R lacking a stretch of 29 amino acids in the ICL3 of D_2L_R. Co-IP results indicate that MOR can heterodimerize with D_2S_R as well, thus proving that the interaction sites for the dimer do not lie in the stretch of 29 amino acids of ICL3, unlike reported for D_1_R-D_2_R heterodimers [38] (Appendix A).

### 3.2. Fluorescently Tagged D_2L_R and MOR are Functionally Expressed on the Plasma Membrane of Hela Cells

Following the demonstration of co-IP, we next wanted to study D_2L_R–MOR heteromerization in living cells, for which we generated fluorescent and luminescent protein-tagged receptors, which could be used to study the interaction either by FRET, BRET, NanoBiT^®^, or live cell imaging. We first validated the functionality and localization of these tagged receptors. To determine whether the fluorescent protein tag affected the plasma membrane localization of D_2L_R or MOR, HeLa cells transfected with MOR-YFP or D_2L_R-mCherry were analyzed using fluorescence microscopy. In both instances, fluorescent YFP or mCherry signals were primarily present at the plasma membrane, although for D_2L_R, a small fraction was observed intracellularly, likely corresponding to the endoplasmic reticulum, from where this receptor traffics (Figure 2). Functionality of the tagged receptors was next evaluated by monitoring their capacity to internalize or stimulate calcium release upon stimulation with their respective agonists. When stimulated with their respective agonists, fluorescently tagged MOR and D_2L_R internalized and calcium release was observed following ligand-mediated stimulation of cells expressing split nanoluciferase-tagged MOR and D_2L_R (Appendix A).

### 3.3. Analysis of MOR-D_2L_R Dimerization by BRET^1^

To verify D_2L_R-MOR interaction in intact living cells, we used a noninvasive saturation BRET^1^ assay. In this assay, HEK 293T cells were transfected with plasmids encoding a constant amount of MOR-*R*Luc or D_2L_R-*R*Luc as donor and increasing amounts of D_2L_R-YFP/MOR-YFP. For cells expressing MOR-*R*Luc, a control was also included where EYFP alone was the acceptor species. When the *R*Luc substrate, *h*-coelenterazine, is added, the energy generated from it excites YFP and emission fluorescence from it is recorded as a BRET signal. The BRET signal increased concomitantly with increasing amounts of the acceptor, until finally reaching saturation, giving rise to a hyperbolic curve, indicative of physical interaction between MOR and D_2L_R (Figure 3A,B). On the other hand, as exemplified for MOR-*R*Luc, the BRET signal was considerably smaller, increasing linearly with increasing amounts of EYFP in the condition where MOR-*R*Luc was coexpressed with EYFP as a negative control. The experiment was performed using both receptors as donor as well as acceptor, thus negating the possibility of an artifact introduced by the tags.

### 3.4. Receptor Interaction Analysis by Functional Complementation of a Split Luciferase

Next, we performed a functional complementation (NanoBiT^®^) assay to detect heterodimerization in living cells. This was done as the tags used in the BRET assay (luciferase and YFP) are both quite large, which may have an impact on the actual behavior of the receptors. To overcome this, tags based on a split nanoluciferase (NanoLuc) were utilized to study protein complementation. The receptors were C-terminally fused to the split fragments (LargeBiT, LgBiT {18kDa}, and SmallBiT, SmBiT {1kDa}) of the luminescence reporter, nanoluciferase. An equal amount of EGFP was cotransfected in every condition and the luminescence was normalized across the different set-ups. Interaction between MOR-LgBiT and D_2L_R-SmBiT results in functional complementation of nanoluciferase, which upon addition of the substrate furimazine results in a luminescent signal. As a negative control, HaloTag fused to SmBiT was used due to the ubiquitous expression of HaloTag inside the cell [39]. The signal produced due to MOR-LgBiT + D_2L_R-SmBiT was approximately 3.5-fold higher than the negative control MOR-LgBiT + HaloTag-SmBiT. Cells expressing only MOR-LgBiT or D_2L_R-LgBiT only yielded a background signal (Figure 4). Interestingly, when switching the nanoluc tags between both receptors (MOR-SmBiT + D_2L_R-LgBiT), no evidence for interaction was obtained, suggesting that a specific configuration is required (Figure 4).

### 3.5. Assessment of MOR-D_2L_R Interaction Using FRET

We further tested the MOR–D_2L_R interaction using FRET, with YFP serving as the donor species and mCherry as acceptor. HeLa cells were used instead of HEK 293T cells, as they are larger and better suited for microscopy purposes. For this, we cotransfected cells with MOR-YFP and either D_2L_R-mCherry or, as a negative control, mCherry-CAAX, whereby the CAAX motif is sufficient to target mCherry to the plasma membrane [40]. To obtain a comparable copy number of the mCherry acceptor molecules, a titration was performed with different concentrations of mCherry-CAAX versus D_2L_R-mCherry, and an optimized amount of mCherry-CAAX was used, which produced approximately the same intensity as D_2L_R-mCherry. The FRET signal in HeLa cells coexpressing MOR-YFP and D_2L_R-mCherry was significantly higher than that in control cells coexpressing MOR-YFP and mCherry-CAAX (Figure 5).

### 3.6. Mobility of MOR is Altered on the Plasma Membrane in the Presence of D_2L_R

Given that the plasma membrane has a certain fluidity, the lateral mobility of membrane proteins is affected by their interaction with neighboring proteins [41,42,43,44]. Hence, FRAP would aid in understanding if mobility of MOR is altered upon interaction with D_2L_R at the plasma membrane. To address this, HeLa cells expressing MOR-YFP with and without D_2L_R-mCherry were used and membrane FRAP analysis was performed by bleaching the MOR-YFP protein at several defined regions on the plasma membrane, followed by monitoring of its recovery. Although the difference in half maximal recoveries between both conditions, i.e., MOR-YFP in the presence and absence of D_2L_R-mCherry, did not reach statistical significance, it was observed that the time for half maximal recovery of MOR-YFP increased from ~77 s when D_2L_R-mCherry was not present to ~85 s when D_2L_R-mCherry was coexpressed (Figure 6). Although this finding should be interpreted cautiously, this slight change in mobility of MOR upon coexpression of D_2L_R lends further support to the presence of an interaction between the two GPCRs.

### 3.7. Heteromerization of MOR and D_2L_R Inhibits the Internalization of MOR after Activation

Studies have shown that heteromers could act as a functional unit that is different from the monomers. Moreover, there have been enormous efforts to improve our understanding of the regulation of MOR by desensitization, internalization, dimerization, etc. With this in mind, we sought to examine the influence of interaction between MOR and D_2L_R on one of the most characteristic responses following activation of GPCRs, i.e., the internalization of MOR following agonist stimulation. For this, we used live cell imaging of HeLa cells transfected with pMOR-YFP with and without pD_2L_R-mCherry. Forty-eight hours post-transfection, cells were washed and imaged live for changes in the distribution of YFP protein over time. Upon stimulation with the MOR specific agonist, DAMGO (7 µM), a robust internalization of MOR-YFP, when present alone, could be recorded from the plasma membrane to the endomembranes within 10 min (Figure 7, black curve). The internalization was quantified by analyzing cytosolic intensity as a function of time (as described in the materials and methods section). The graphs depict the cytosolic intensity of a population of cells after stimulation, normalized to their basal condition in the absence of the agonist. In the presence of D_2L_R-mCherry, a marked reduction in the internalization of MOR upon stimulation with its agonist DAMGO was recorded, thus suggesting an influence of heterodimerization on the internalization of MOR. In all experiments, comparable expression intensities of MOR-YFP were recorded to avoid expression-related artifacts. In cells expressing only D_2L_R-EGFP that were stimulated with DAMGO, no D_2L_R-EGFP internalization could be observed, confirming that DAMGO does not act via the D_2_R (Appendix A).

## 4. Discussion

Several studies have demonstrated the presence of homo-, hetero-, and even higher order oligomers of GPCRs in cells and in tissues [45,46], but the functional relevance of these interactions has remained unexplored in many instances. In a study performed by Dai et al. [35], an increased expression of neuronal D_2_R in the spinal dorsal horns of mice was observed upon chronic treatment with morphine. Immunofluorescence studies revealed that D_2_R colocalized with MOR in the spinal cords of mice, and both receptors could be coimmunoprecipitated. Upon chronic morphine treatment, this interaction increased, while blockade of D_2_R with sulpiride disrupted the interaction and also attenuated morphine tolerance, suggesting that increased MOR–D_2_R interaction may play a role in chronic morphine tolerance. Our study is in agreement with the findings of Dai et al. [35] and lends further support for the possible interaction between MOR and D_2_R, as supported by various approaches such as co-IP, BRET^1^, NanoBiT^®^, and FRET. Furthermore, we observed an influence of dimerization between D_2L_R and MOR on DAMGO-mediated internalization of MOR.

The mu opioid receptor is of significant importance due to its pivotal role in mediating the effects of analgesics like morphine and owing to its undesirable side effects including addiction, dependence, etc. In 1999, Jordan et al. [47], showed for the first time that another member of the opioid receptor family, the delta opioid receptor (DOR) could heterodimerize with the kappa opioid receptor. Since then, several reports have been published on the dimerization of MOR with DOR [31,48,49], and with other GPCRs [32,50,51,52], thereby starting a new dimension in the field of analgesia. On one hand, efforts are constantly being dedicated to developing new ligands that are agonists of MOR, but do not have the undesired side effects. On the other hand, fundamental research is being carried out to understand the signaling of MOR, so that it could be modulated with ligands. E.g., Qian et al. [53] recently reported on the usage of heterobivalent ligands based on agonists and antagonists of MOR and D_2_-likeR to act as pharmacological tools that should allow us to gain more insight into heteromers.

In the present study, we have obtained strong evidence for the presence of D_2_R and MOR heterocomplexes, using two different cell lines and a variety of chimeric receptors fused to different tags for studying protein–protein interactions. All the evaluated chimeric receptors were found to be active through their ability to initiate calcium signaling [17] or by their capacity to internalize upon stimulation with an agonist.

Co-IP was used to check the heterodimerization and the results indicated the existence of MOR–D_2_R heterocomplexes in cotransfected cells (Figure 1). To address the question about specificity of the interaction, lysates of single receptor transfected cells were mixed, and the lack of co-IP indicated that the receptors physically interact with each other within a given cell (Appendix A).

To clearly illuminate the nature of the dimerization, we used an array of techniques using living cells expressing receptor fusion constructs, one such technique being BRET^1^. We observed strong BRET signals between D_2_R and MOR, implying that the two receptors are within a distance of 10 nm, and the experiment was performed in both ways, to strengthen our hypothesis. Since overexpression of receptors may lead to random collisions and consequently produce a false signal, we have also included a negative control wherein the same saturation BRET^1^ assay was performed with receptor-*R*Luc and increasing concentrations of EYFP. The linear nature of the negative control, as compared to a hyperbolic curve obtained with co-expressed receptors, showed that the receptors interact with each other (Figure 3).

A recently developed luminescence complementation-based approach, developed by Promega, known as “Nanoluciferase Binary Technology”/NanoBiT^®^, was also used to study the interaction between D_2L_R and MOR. The signal obtained for the interaction (D_2L_R-SmBiT + MOR-LgBiT) was approximately 3.5-fold higher than that of the negative control (Halotag-SmBiT + MOR-LgBiT) (Figure 4). Also, the background signal obtained with receptor-LgBiT fusions alone was low. An interesting observation made here was that when the interaction was studied with the tags switched (MOR-SmBiT + D_2L_R-LgBiT), only a low signal was obtained, indicating that a specific configuration seems to be required for functional complementation of the luminescent protein. More specifically, the configuration in which SmBiT and LgBiT are fused to the C-termini of MOR and D_2L_R, respectively, may not allow functional complementation of the split fragments of nanoluciferase because of sterical hindrance or misorientation.

We also evaluated the formation of the heterodimer in another cell line, HeLa, using FRET. Different variants of FRET have been used to study GPCR dimerization. Recently, Niewiarowska-Sendo et al. [54] reported the formation of a functional dimer between the bradykinin receptor and the dopamine D_2L_R using fluorescence lifetime imaging microscopy-based fluorescence resonance energy transfer (FLIM-FRET) technique. In our experiments, in which we chose MOR-YFP as the donor and D_2L_R-mCherry as the acceptor, a significantly higher signal was obtained than in the negative control, acceptor mCherry-CAAX, providing a strong evidence for dimer formation (Figure 5).

In 2009, Dorsch et al. [55] applied dual color FRAP to answer the question on GPCR oligomerization. With the help of polyclonal antibodies against YFP, they immobilized the YFP-tagged receptor, while the other interacting partner was tagged to CFP. By means of dual color FRAP, they observed that the β_1_-adrenergic receptor (β_1_-AR) formed monomers that are unstable, in contrast to the β_2_-adrenergic receptor (β_2_-AR), which formed stable higher-order oligomers. The dimerization event in Class A GPCRs is more complicated than in other classes since there are multiple interfaces and owing to the fact that the interactions are transient in nature [56]. For D_2_R–D_2_R interactions, this was recently supported by modelling studies [23] and by studies indicating a dimer lifetime of only 68 ms [57,58]. In our study, we performed FRAP by bleaching MOR-YFP at several regions on the plasma membrane with or without coexpressed D_2L_R-mCherry. Although statistically not significant, a slight decrease in mobility of MOR-YFP in the presence of D_2L_R-mCherry was found, which supports the hypothesis that the two GPCRs interact with each other (Figure 6).

In recent years, there have been substantial efforts to understand the molecular regulation of desensitization and internalization of MOR [4,59,60,61,62,63]. Studies have shown that the MOR agonists fentanyl and DAMGO are more efficacious at recruiting β-arrestin to the receptor [64,65], subsequently causing internalization of the receptor, when compared to morphine, which shows delayed recruitment of β-arrestin and a slower internalization [66,67,68]. It is the recruitment of β-arrestin that has been associated with the unwanted effects of opioids [11,69,70,71].

With this in mind, we sought to study the effect of heterodimerization of MOR-D_2L_R on the internalization characteristics of MOR. For this, time-lapse imaging is one of the best suited techniques to understand the spatial and temporal aspects of internalization of the receptor. Here, the dynamics of MOR-YFP internalization were studied in HeLa cells, with and without co-expression of D_2L_R-mCherry. MOR-YFP showed robust internalization when stimulated with DAMGO, in agreement with the literature. Upon co-expression of D_2L_R, and when stimulated with DAMGO, we noticed that the internalization of MOR-YFP slowed down (Figure 7), thus suggesting that the heterodimerization may have a role in modulating internalization of the interacting partner.

In a recent study by Dai et al. [21], it was demonstrated that *l*-CDL could attenuate morphine tolerance in rats with chronic bone cancer pain. This compound proved to be an antagonist of D_2_R as it inhibited dopamine-induced calcium release in CHO-K1 cells expressing D_2_R. A similar effect, as observed in their previous study [35], was noticed upon coadministration of *l*-CDL with morphine, and the effect was reversed upon addition of quinpirole, a D_2_R agonist in mice. The findings were reconfirmed when they observed diminished tolerance upon administration of D_2_R siRNA, which silenced the D_2_R gene. Chronic morphine tolerance upregulated the expression of β-arrestin 2, which decreased both upon addition of D_2_R-siRNA and *l*-CDL. Both the phosphoinositide 3-kinase (PI3K)/Akt pathway and the mitogen-activated protein kinase (MAPK) pathway are involved in the development of morphine tolerance, and both D_2_R siRNA and *l*-CDL reduced the levels of key phosphorylated proteins of both pathways.

Another study provided evidence for an enhanced antinociceptive effect of the MOR agonist DAMGO, when coadministered with quinpirole [22]. In intact animals, this synergistic effect of antinociception was only seen in mechanonociceptive tests. However, in neuropathic or inflammatory models of pain, quinpirole enhanced antinociception in both mechanonociceptive and thermonociceptive tests. Also, the enhanced antinociceptive effect observed by coadministration of subeffective doses of DAMGO and quinpirole was driven by D_2_R because this effect was abolished by administration of the highly selective D_2_-like receptor antagonist raclopride.

Although these studies clearly suggest an interplay between D_2_R and MOR, the link between D_2_R and opioid analgesia, as well as tolerance, still remains unclear as there are reports for both D_2_R antagonists and D_2_R agonists to attenuate morphine tolerance [21,35].

## 5. Conclusions

The data presented here and obtained using a combination of biochemical techniques, co-IP, BRET^1^, NanoBiT^®^, and FRET, shed light on the heterodimerization between MOR and D_2_R. In addition, the change in internalization of MOR in the presence of D_2_R suggests that these receptors functionally interact with each other. Future studies to evaluate the influence of heterodimerization of MOR on G protein signaling following receptor activation may lead to a better understanding on whether the antinociceptive effects of MOR agonists are affected. Based on current findings and future experiments, novel therapies targeting the heterodimers could be designed. Taken together, this study improves our understanding of the complex cellular network that influences signaling through a GPCR. Much remains to be explored about the kind of downstream signaling that is altered due to the heterodimerization process requiring following research, using both in cellulo as well as in vivo approaches.

## Figures and Tables

**Figure 1 biomolecules-09-00368-f001:**
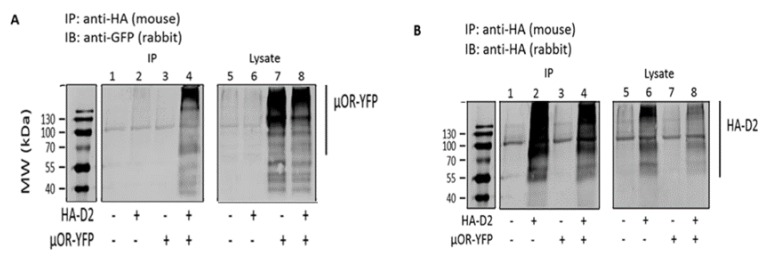
Assessment of D_2_R receptor dimerization with mu opioid receptor (MOR) using coimmunoprecipitation. HEK 293T cells were transiently transfected with pHA-D_2L_R, pMOR-YFP, or both. After 48 h, the cells were lysed, and an aliquot of the lysates was subjected to SDS-PAGE followed by immunoblotting with anti-HA or anti-GFP. The rest of the lysates was subjected to immunoprecipitation (IP) with anti-HA. Coimmunoprecipitation of D_2L_R and MOR was detected by immunoblotting with anti-GFP (**A**) and anti-HA (**B**).

**Figure 2 biomolecules-09-00368-f002:**
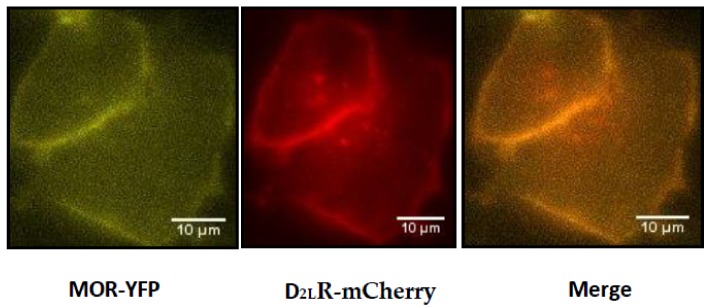
Localization analysis of the tagged receptors in live cells. HeLa cells were transiently transfected with pMOR-YFP (yellow) and pD_2L_R-mCherry (red) and subsequently (48 h post-transfection), their localization was examined by epifluorescence microscopy.

**Figure 3 biomolecules-09-00368-f003:**
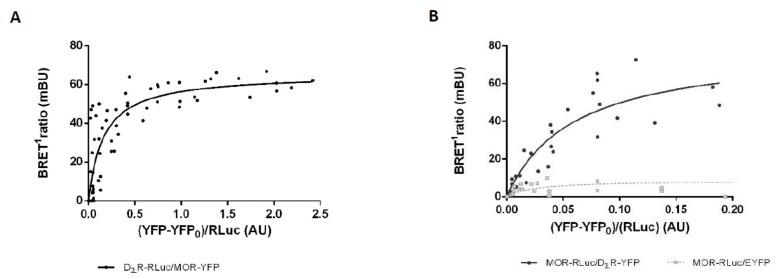
Assessment of D_2L_R receptor dimerization with MOR using saturation bioluminescence resonance energy transfer (BRET^1^) assay. Saturation BRET^1^ assay was performed in HEK 293T cells transfected with a fixed amount of the donor plasmid: D_2L_R-*R*Luc (**A**) or MOR-*R*Luc (**B**) and increasing amounts of the acceptor plasmid: MOR-YFP (**A**)/D_2L_R-YFP (**B**)/EYFP (**B**). The data represent three–four independent experiments, fitted using nonlinear regression equation, assuming a single binding site.

**Figure 4 biomolecules-09-00368-f004:**
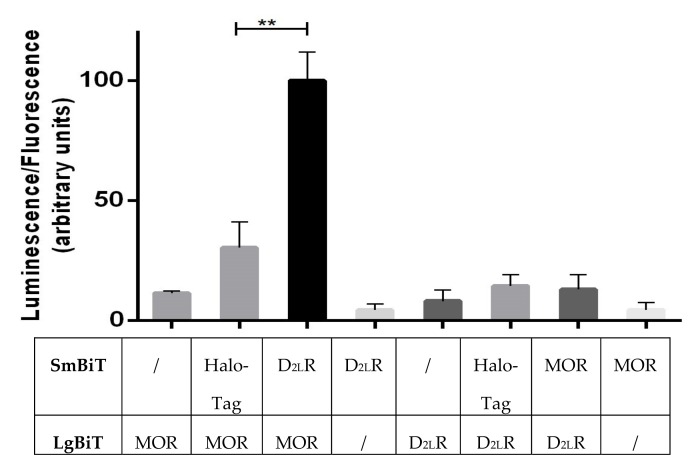
Analysis of D_2L_R interaction with MOR using NanoBiT^®^. HEK 293T cells were transfected with SmBiT and LgBiT-tagged receptor constructs as indicated above. The luminescent signal in cells cotransfected with D_2L_R fused to SmBiT and MOR fused to LgBiT was significantly higher (~3.5 fold) than the signal of the negative control, cells coexpressing Halotag-SmBiT and MOR-LgBiT. Three independent experiments were performed, and the results were expressed as mean ± SD (** *p* < 0.01).

**Figure 5 biomolecules-09-00368-f005:**
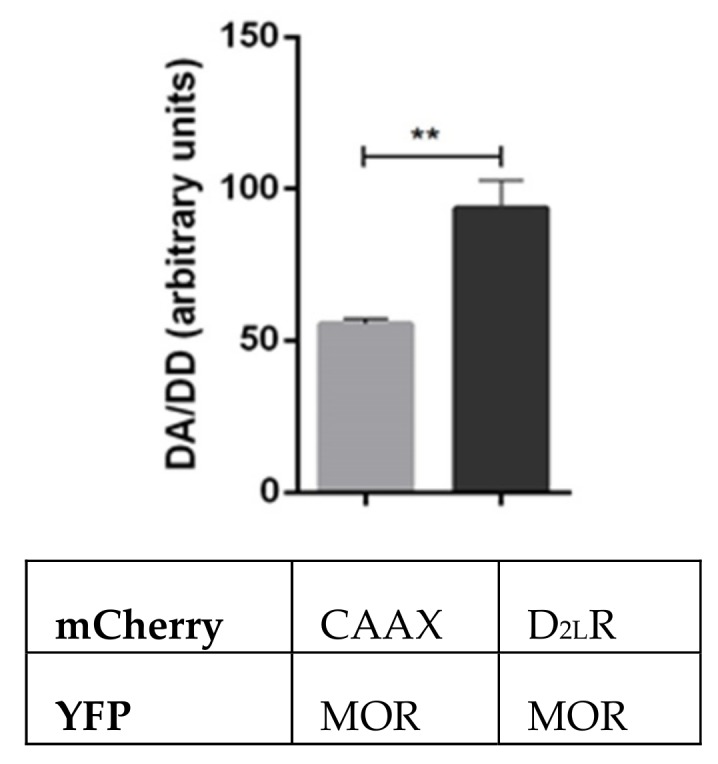
Analysis of D_2L_R dimerization with MOR using Fӧrster Resonance Energy Transfer (FRET). HeLa cells were transfected with pMOR-YFP, and either pD_2L_R-mCherry or pmCherry-CAAX. Donor–acceptor (DA)/ donor–donor (DD) ratios were determined under identical imaging conditions as described in the materials and methods section. Three independent experiments were performed, and results were expressed as mean ± SD (** *p* < 0.01).

**Figure 6 biomolecules-09-00368-f006:**
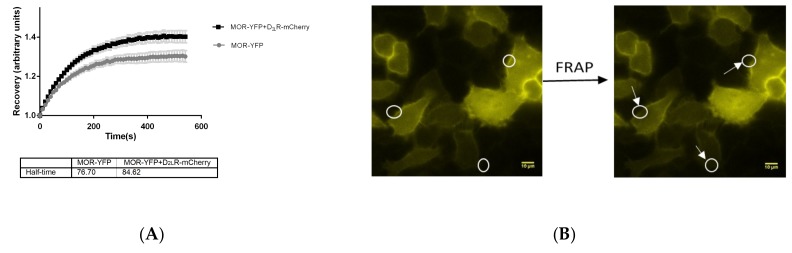
Analysis of interaction-mediated mobility change of MOR upon coexpression of D_2L_R by FRAP analysis. HeLa cells transfected with pMOR-YFP with and without pD_2L_R-mCherry were used for FRAP measurements, as described in the materials and methods section. (**A**) The plot depicts mean relative fluorescence intensity normalized to prebleach intensity as a function of time. The plot is an average for a population of approximately 50–60 cells from three–four independent experiments. The t_1/2_ value refers to the time taken for half maximal recovery after photobleaching. (**B**) Representative images of a FRAP experiment. Several regions on the plasma membrane were photobleached and FRAP was monitored for 9 min with 10 s interval between every image.

**Figure 7 biomolecules-09-00368-f007:**
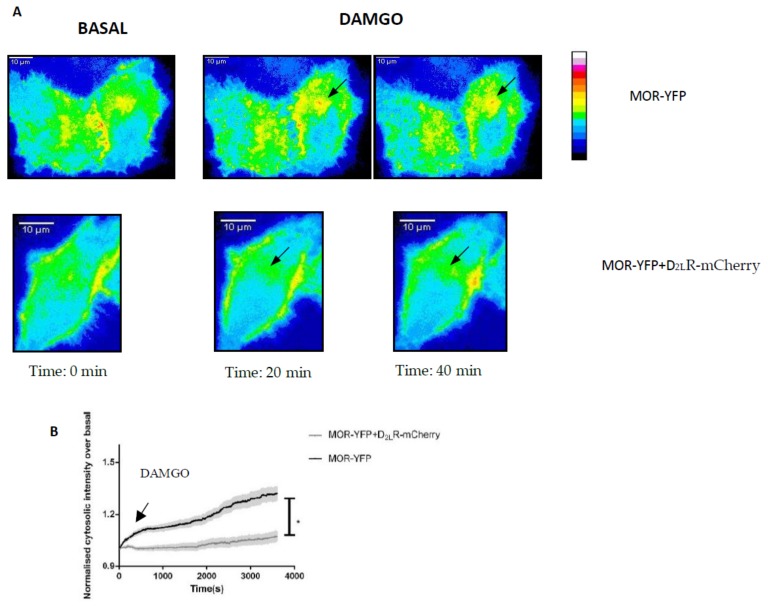
Analysis of internalization of MOR upon stimulation with DAMGO in the presence or absence of D_2L_R by fluorescence live cell imaging. (**A**) HeLa cells transfected with pMOR-YFP with or without pD_2L_R-mCherry were used for fluorescence imaging as described in the materials and methods section. The images were captured for 1 h, with 10 s between successive images. At the 10th frame (100 s), DAMGO (7 µM) was injected using a syringe pump. (**B**) The plot depicts cytosolic fluorescence intensity normalized to the intensity before stimulation as a function of time. Approximately 50–60 cells were analyzed for each system. The results were expressed as mean ± standard error of mean (SEM) (* *p* < 0.05).

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
