# Peer review of "Heterodimerization of Mu Opioid Receptor Protomer with Dopamine D2 Receptor Modulates Agonist-Induced Internalization of Mu Opioid Receptor"

_biomolecules, 2019, doi:10.3390/biom9080368_

Round 1

Reviewer 1 Report

This manuscript uses a variety of tools to test whether or not dopamine D2 and Mu opioid receptors interact forming heterodimers, and how this protein-protein interaction changes the traffic of Mu receptors from the membrane to intracellular compartments. The manuscript is very well written and clear. The authors have extensive experience in the technologies used and presented in this manuscript. The conclusion is well supported with the data presented.

One minor comment, I think the discussion could be improved. Significant length of the discussion is focused in supporting evidences about the existence of heterodimers or oligomeric receptors complexes, very little discussion is found regarding the functional significance D2-Mu receptors hererodimers. The dopamine-opioid interplay in physiological events, this is particularly interesting given that D2 receptors inhibits the internalization of Mu receptors. There are pharmacological evidences of synergistic antinociceptive effect between D2 and Mu agonists [Mercado-Reyes et al., 2019], changes in morphine tolerance by the blockade of D2 dopamine receptors, have been also described [Dai et al., 2016]. How the results presented in this manuscript are in agreement or not, with the physiological and pharmacological evidences is missing in the discussion.

Another minor comment, is not clear the explanation in line 408 page 11 of why when using NanoBiT the signal obtained for the tag D2LR-SmBiT+MOR-LgBiT was higher than the switched tag (MOR-SmBiT+D2LR-LgBiT), please elaborate why you mean by "specific conformation".

Reviewer 2 Report

The manuscript is interesting.

I have only minor comments.

1.The manuscript would benefit from inclusion of introducing/bridging sentences between the individual parts of the "Results" that explain the logical order and rationale for the experiments

2. Please add the IC50.

In the conclusions , the Authors should highlight the possible clinical significance of their findings
